# Cluster randomised controlled trial to assess a tailored intervention to reduce antibiotic prescribing in rural China: study protocol

Wenjuan Cong [1], Jing Chai [2], Linhai Zhao,[2] Christie Cabral,[1] Lucy Yardley,[3] Guiqing Lily Yao,[4] Tingting Zhang [1], Jing Cheng,[2] XingRong Shen,[2] Rong Liu,[2] Paul Little [5], Beth Stuart,[5] Xiaowen Hu,[2] Ye-Huan Sun,[6] Isabel Oliver,[7] Bo Zheng,[8] Helen Lambert,[1] DeBin Wang [2]

WC and JC contributed equally. CC and DW contributed equally.

For numbered affiliations see end of article.

**Correspondence to**
Dr DeBin Wang;
dbwang@vip.sina.com

## ABSTRACT

**Introduction** Up to 80% of patients with respiratory tract infections (RTI) attending healthcare facilities in rural areas of China are prescribed antibiotics, many of which are unnecessary. Since 2009, China has implemented several policies to try to reduce inappropriate antibiotic use; however, antibiotic prescribing remains high in rural health facilities.

**Methods and analysis** A cluster randomised controlled trial will be carried out to estimate the effectiveness and cost effectiveness of a complex intervention in reducing antibiotic prescribing at township health centres in Anhui Province, China. 40 Township health centres will be randomised at a 1:1 ratio to the intervention or usual care arms. In the intervention group, practitioners will receive an intervention comprising: (1) training to support appropriate antibiotic prescribing for RTI, (2) a computer-based treatment decision support system, (3) virtual peer support, (4) a leaflet for patients and (5) a letter of commitment to optimise antibiotic use to display in their clinic. The primary outcome is the percentage of antibiotics (intravenous and oral) prescribed for RTI patients. Secondary outcomes include patient symptom severity and duration, recovery status, satisfaction, antibiotic consumption. A full economic evaluation will be conducted within the trial period. Costs and savings for both clinics and patients will be considered and quality of life will be measured by EuroQoL (EQ-5D-5L). A qualitative process evaluation will explore practitioner and patient views and experiences of trial processes, intervention fidelity and acceptability, and barriers and facilitators to implementation.

**Ethics and dissemination** Ethical approval was obtained from the Biomedical Research Ethics Committee of Anhui Medical University (Ref: 20180259); the study has undergone due diligence checks and is registered at the University of Bristol (Ref: 2020-3137). Research findings will be disseminated to stakeholders through conferences and peer-reviewed journals in China, the UK and internationally.

**Trial registration number** ISRCTN30652037.

## Strengths and limitations of this study

► This is a large cluster randomised trial to determine whether the intervention is effective in reducing antibiotic prescribing for patients with respiratory tract infection in township health centres compared with usual care.

► This study will also evaluate the cost-effectiveness of an intervention to reduce antibiotic prescribing in rural areas.

► The quantitative and qualitative research in the trial could also provide recommendations on how to reduce antibiotics prescribing at village clinic level.

► The use of direct observation will ensure accurate assessment of antibiotic prescribing in both baseline and intervention phases.

► The study, though selected to be representative of rural populations in China, includes 10 counties from 4 regions across the Anhui Province, which may have limited generalisability.

micro-organisms, including bacteria, which is driven by high rates of antibiotic use and is a pressing problem for global health and development.[1–3] If resistance continues to increase, common bacterial infections will eventually become untreatable.[4 5]

Since 2009, China has implemented the national essential medicines scheme and zero-mark-up policies to try to reduce inappropriate antibiotic use.[6 7] In 2012, the Chinese Ministry of Health issued a regulation to limit antibiotic prescribing to 20% of outpatient prescriptions in all patients.[8] However, the antibiotic prescribing rate in healthcare facilities in less developed or rural areas remains as high as 80%.[9–14]

In China, patients can choose to go to the outpatient department of any health facility without a referral letter,[15] however, rural

## INTRODUCTION

Antimicrobial resistance (AMR) specifically refers to the evolution of drug-resistant

residents mainly seek healthcare from township health centres or village clinics, the two lowest-level health facilities in China's rural healthcare system. If patients develop severe symptoms, they are more likely to go directly to the outpatient clinic in the county hospital.[15 16] Many patients attend village clinics and township health centres with respiratory tract infections (RTIs) such as common cold, sore throat, and acute bronchitis and are frequently prescribed antibiotics even though most cases are likely to be viral and antibiotics are therefore unnecessary[17–20] Our previous study found that the antibiotic prescribing rate for RTI patients was up to 85% in rural Anhui Province.[21] In addition to the implications for AMR, there is a cost to patients[22 23] for extra health resource utilisation (consultation fees, medication costs, loss of productivity costs), most of which would not be reimbursed through national medical insurance schemes. Therefore, reducing antibiotic prescribing in village clinics and township health centres is likely to be both safe and economically beneficial to patients. Since the start of the COVID-19 pandemic, village clinics have periodically been prohibited from providing treatment to RTI patients; therefore, this trial focuses on provision of care in township health centres.

Township health centres in China are staffed with both qualified doctors and some health practitioners who are usually referred to as 'village practitioners' but would not have a degree in medicine. Village practitioners are required to undergo basic training in healthcare provided by the local healthcare authority issued with a medical practice licence. Antibiotic prescribing in township health centres is influenced by uncertainty about which RTI need antibiotics and by widespread beliefs that common infections should be treated with antibiotics.[15] Practitioners are also concerned about losing patients and believe that prescribing antibiotics helps maintain good relationships with their patients. Public campaigns on antimicrobial stewardship (AMS)[23–25] are very limited in rural areas and there is little technical support for village and township practitioners from senior clinical consultants and healthcare authorities, which makes it difficult for them to change norms of high antibiotic use.

This study will aim to determine whether the intervention safely reduces antibiotic prescribing for patients with RTI in township health centres. The study will provide evidence of effectiveness, value for money and barriers and facilitators to implementation that can inform policy and clinical practice. If pandemic conditions permit, the study will also pilot implementation of the same intervention in a limited number of village clinics to demonstrate proof of concept in lower level health facilities.

## METHODS AND ANALYSIS
### Aims and objectives
This study aims to evaluate the effectiveness of a complex intervention aimed at healthcare practitioners and patients at township health centres, to reduce antibiotic prescribing for RTIs compared with usual care in township health centres.

The primary outcome will be the percentage of antibiotics (intravenous and oral) prescribed for RTI patients.

Secondary objectives are to:
► Estimate the cost-effectiveness of the intervention and the cost implications for both township health centres and patients.
► Evaluate the impact of reduced antibiotic use on patient satisfaction with care.
► Evaluate the impact of reduced antibiotic use on patient symptom severity, duration and recovery rates.
► Investigate patient antibiotic consumption rates (prescribed or obtained from another source).
► Understand practitioner and patient views of the intervention and barriers and facilitators to implementation.

### Trial design
This is a pragmatic, two-armed (intervention vs usual care) trial with an embedded cost-effectiveness and qualitative evaluations, with randomisation at township health centre level, using observed antibiotic prescribing to assess effectiveness.

### Participants and recruitment procedures

### Selection of township health centres
We originally started the trial recruitment at village clinic level. However, in May 2021, there was a small local COVID-19 outbreak in Anhui Province, since when village clinics have not been allowed to treat RTI patients in Anhui and our trial recruitment had to be postponed. Currently, patients with suspected COVID-19 symptoms (specifically patients with a temperature of more than 37.3°) must be referred to specialist COVID-19 clinics or specialist COVID-19 hospitals for treatment. We did some preliminary fieldwork in June 2021 and found it is still very difficult to recruit RTI patients at village clinic level and the research team assessed that this situation may not change within the next 12 months. However, we found that township health centres are still able to treat RTI patients regularly. In order to complete recruitment and maintain trial feasibility, we therefore decide to change the recruitment sites from village clinics to township health centres.

We will recruit 40 township health centres from 10 counties (4 township health centres per county), those counties will be randomly selected from 4 regions geographically dispersed across Anhui Province. We will randomly recruit 6 counties from two regions in densely populated North Anhui and four counties from another two regions from sparsely populated South Anhui. We will seek support from local county health authorities during the recruitment of township health centres. They will provide a list of townships in their area and the research team will randomly select four township health centres. Local county health authorities will screen the selected

townships and may exclude selected townships, for which a reason will be recorded.

## Recruitment of practitioners

The recruitment of health practitioners will be fair. Once a township health centre is recruited and has given permission for its staff to participate, all the practitioners working in the recruited township health centres will be invited to participate in the study. Once practitioners are identified and recruited, informed consent (online supplemental appendix 1) will be obtained immediately.

Exclusion criteria

1. Practitioners who do not have basic computer skills.
2. Practitioners who are not involved in clinical diagnosis and medicine prescribing.

The background information of participating township health centres and practitioners will be collected. A senior practitioner in the recruited township health centre will complete a clinic background information questionnaire (online supplemental appendix 2); and all practitioners working in the clinic will give their demographic data in

a practitioner background information questionnaire (online supplemental appendix 3).

A baseline audit of 400 RTI patients across the 40 township health centres will be conducted (10 RTI patients for each township health centre). This baseline audit will only collect primary outcome data, including the antibiotic prescribing rates at each clinic. Then all the township health centres will be randomised at a 1:1 ratio to usual care group versus intervention group, using the baseline antibiotic prescribing rates to balance high and low prescribing clinics between the arms. Each group will have 20 township health centres (details see figure 1).

## Recruitment of patients

All patients with RTIs who seek healthcare from the recruited health centres will be screened for eligibility and invited to participate in this study. Researchers from Anhui Medical University will undertake patient recruitment on-site until 30 participants have been recruited at each township health centre. Informed consent (online

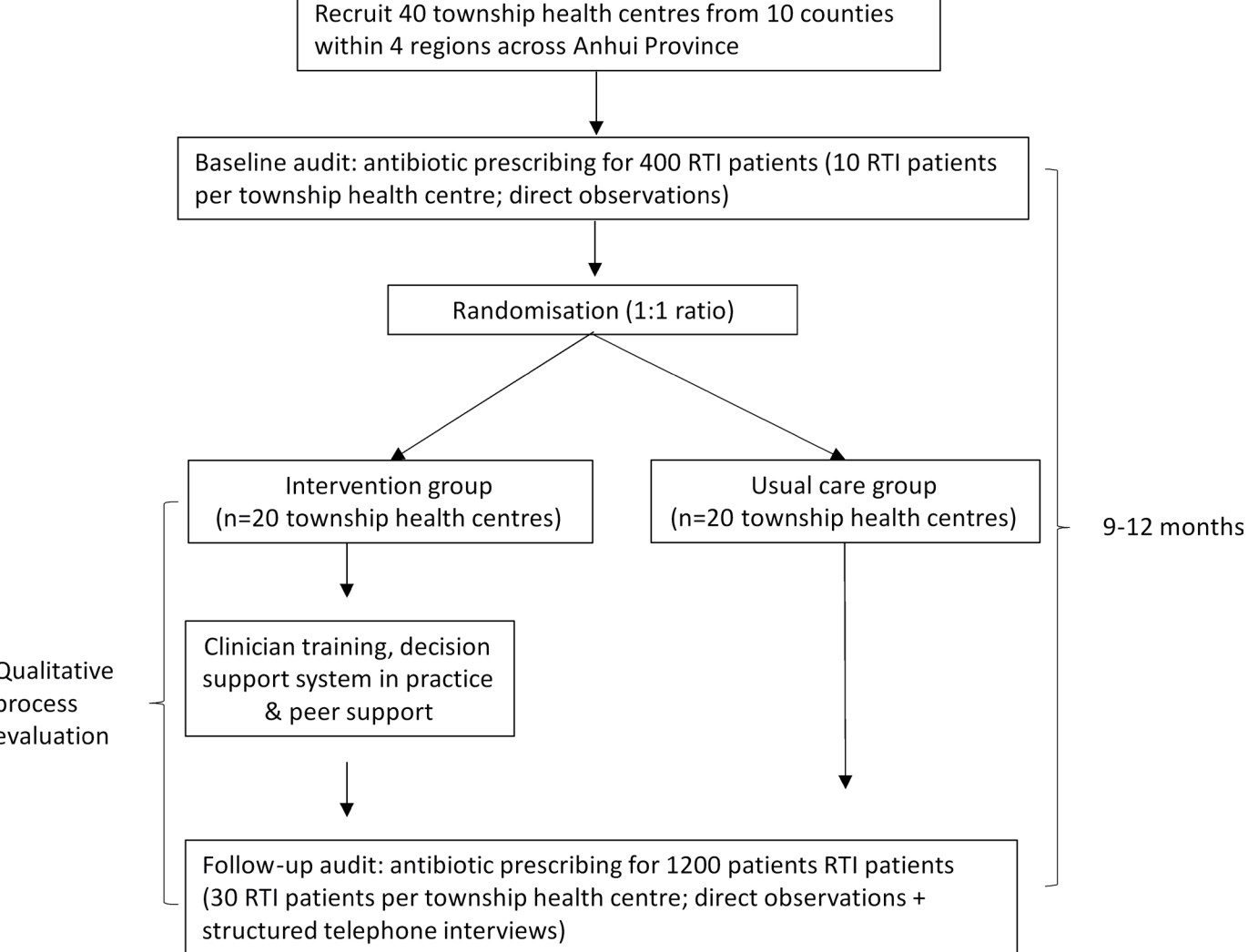

**Figure 1** Trial flow chart. RTI, respiratory tract infection.

supplemental appendix 4) will be obtained from all the patients who agree to participate in this study.

Currently, patients with suspected COVID-19 symptoms must be referred to specialist hospitals and township health centres are not expected to see suspected COVID-19 patients who present for treatment. However, some of these patients test negative for COVID-19 and are then referred back to their local clinic for treatment for their non-Covid RTI. These patients will be eligible for inclusion in our study.

### Eligibility Criteria
#### Inclusion criteria
1. Adult (aged ≥18 years old).
2. Patient who is diagnosed with an RTIs in the recruitment consultation: blocked/runny nose, coughing with or without sputum, dry/sore throat, breathing problems, fever, ear inflammation (blocked ear, tinnitus, ear discharges, earache).

#### Exclusion criteria
1. Patients who have previously sought treatment for this illness from a normal clinic (not a specialist COVID-19 clinic).
2. Patients who are unable to provide fully informed consent (eg, dementia, psychosis or severe depression).
3. Patients who are pregnant.

Patients without bacterial infections, immunocompromised patients and patients with non-infective conditions are not excluded from this study if they present with any of the symptoms listed in the inclusion criteria. Patients who previously sought treatment for the same episode of illness in other health services (outpatient clinic in the county hospitals, etc) will be excluded, simply because the treatment sought elsewhere may bring interference or contamination to our study findings by having previously obtained antibiotics elsewhere.

### Randomisation, allocation and blinding
All the recruited township health centres will be stratified by above or below the average baseline antibiotic prescribing rate and randomised, in blocks of 4 and 6, at a 1:1 ratio to the intervention or usual care group. Antibiotic prescribing rate (%) of each township health centre will be measured by dividing the total number of RTI patients who are prescribed antibiotics (intravenous and oral) against the total number of RTI patients who are present to each township health centre.

All participating health centres will be allocated to a trial arm at the same time, after completion of baseline data collection. Due to the nature of the intervention, it is not possible to blind practitioners to their allocation to either intervention or usual care group. However, the statisticians conducting the analysis will remain blind to allocation until quantitative analysis is complete. The trial randomisation and blinding will be conducted by an independent statistician.

### Intervention
The complex intervention comprises training, peer support, a computerised decision aid, a public commitment letter and a patient leaflet. These were codesigned with practitioners and patients using the person-based approach method for intervention development.[26]

We optimised the intervention tools using qualitative research to identify barriers to acceptability, feasibility and engagement that need to be addressed. We conducted four rounds of 'think-aloud' interviews with 37 practitioners (15 from township health centres and 22 from village clinics) and 17 Patients in rural Anhui. Think-aloud interviews obtained detailed user views by asking practitioners and patients to view contents of our intervention tools (training materials for practitioners, peer support groups, computer-based treatment decision support system, patient leaflet). Findings were immediately tabulated and analysed to identify changes required, modified materials used for the next round of think-aloud interviews and the process repeated until participant feedback indicated no further important, feasible and appropriate changes were needed. Through this process, we created a feasible and generalisable intervention to reduce antibiotic prescribing tailored for the rural Chinese context. As an important element of the intervention, our computer-based treatment decision support system is adapted from a previous project where we set up a web-based just-in-time information system, and our colleagues had already tested the feasibility and acceptance of the system with a pilot study.[27] Additionally, we organised several trial steering group meetings where the intervention tools and its feasibility and generalisability were intensively discussed before we launched the trial.

Intervention practitioners will receive training delivered face to face in one session by a senior clinician from a tertiary hospital. The training covers: (1) evidence about AMR in China and the link to their own practice; (2) addresses key concerns and misapprehensions; (3) details the current Chinese clinical guidelines on treatment of RTIs; (4) explains how to use the computer-based treatment decision support aid and (5) outlines communication strategies to improve patient acceptance of a non-antibiotic treatment decision.

Peer support groups will be established to connect the practitioners to others in the intervention group in order to provide peer support to each other during implementation. At the end of the training session intervention practitioners will be asked to organise themselves into peer support groups with a few practitioners in each and identify a lead for each peer support group. Peer support groups will meet via a widely used social media app in China (WeChat). The peer support group leader will schedule two structured Wechat meetings within a month after the training session. The group leader will be provided with a meeting guide to ensure a structured discussion and be responsible for setting up structured Wechat meetings of the peer support group, during which each member of the group will share their experience of

applying the intervention (case discussion), discuss any problems and support each other to find solutions. Practitioners will be encouraged to contact each other outside of the scheduled meetings to ask for and provide support to each other. There will be another technical support group that will include the trial team and all intervention practitioners to provide technical assistance with intervention implementation. The senior clinician and two junior clinicians will be part of this group and be able to provide general guidance and reassurance that the intervention follows National Chinese prescribing guidance but will not provide specific clinical advice on cases.

A computer-based treatment decision support system will be installed on computers in consulting rooms in the intervention clinics. This will allow practitioners to record the patient's symptoms and the practitioner's diagnosis and will then provide the appropriate treatment guidance from the Chinese national guidelines. The treatment decision support system has been adapted from our previous project.[27] It contains the following sections (online supplemental appendix 5): (1) patient eligibility criteria; (2) patient symptoms: allow practitioners to record what symptoms patients are suffering from; (3) diagnosis, patient reassurance and prognosis: allow practitioners to record their diagnosis, and provide appropriate patient reassurance and prognosis for each type of RTI patient; (4) treatment recommendations for each type of RTI patient from Chinese National Guidelines for Treating Acute Respiratory Tract Infections; (5) Table of symptomatic treatment options for RTI patients (also from Chinese National Guidelines); and (6) Safety netting advice (when to come back to see the doctor) (also from the Chinese National Guidelines).

Practitioners will be required to sign a public declaration letter stating that they are committed to reducing unnecessary antibiotic prescribing and display this in a public area of their clinic.

Township health centres will be provided with a leaflet that can be given to RTI patients. This leaflet will address common patient beliefs and concerns about RTI, address common misapprehensions that can lead to an expectation of antibiotic treatment, provides advice on non-antibiotic symptomatic treatment options and safety-netting.

### Data collection methods

Baseline data will be collected for 400 patients with RTI at the participating township health centres. Follow-up data collection will be collected for 1200 patients at the participating township health centres after implementation of the intervention.

In all recruited township health centre, direct observations will be carried out by a researcher from Anhui Medical University following the doctor questionnaire (online supplemental appendix 6) adapted from our previous study.[16] Observation will focus on the adult RTI clinical consultations and record patient symptoms, practitioner's diagnosis & treatment, sickness severity,

any antibiotics or other medicines prescribed and mode of administration (intravenous or oral), name, dosage and quantity. Immediately after the consultation, the researcher will fill in a patient questionnaire (online supplemental appendix 7) by asking questions of patients directly. This questionnaire is also adapted from the previous study[16] and includes questions about patient demographic characteristics (gender, age, education level, residential area and medical insurance records), illness duration, symptom severity and patient satisfaction with the consultation; the latter two questions will be scored using a 10-point scale.

The researcher will conduct telephone interviews with recruited RTI patients at 7, 14, 21 days after their consultation. Structured questionnaires will be completed for each patient by the researcher during the telephone interview, with different secondary outcome data collected at different time points (see table 1). Illness severity (scored on a 10-point scale) and symptom duration will be recorded at each time point. Patient attitude, understanding of the patient leaflet and beliefs will be recorded once at 7 days after consultation (online supplemental appendix 8). Patient continuing illness severity and recovery situation (when they feel better and when they feel they could get back to normal) will be checked at 7 days (online supplemental appendix 8, and 14 days after consultation (online supplemental appendix 9). Patient consumption of antibiotics (including name, days, dosage and quantity) will be recorded at 7, 14 and 21 days after consultation (online supplemental appendix 10). Through checking with patient's actual consumption of antibiotics (how many days they have taken the antibiotics, dosage etc) at different periods against the prescribed quantity of antibiotics (if applicable), patient adherence to follow practitioner's advice (%) could be estimated. Other healthcare use (clinic visits or consumption of other treatment) will be recorded at 21 days after consultation. Patient out-of-pocket spending due to this illness, including purchase of medication, fees for clinic attendance, days off-work, will be recorded at 21 days after consultation. Patient quality of life will be measured using the EQ5D-5L during the initial consultation, at 7, 14 and 21 days after consultation. (Use of paper interview administration mode of EQ5D-5L in Chinese version has been authorised by EuroQol Group (registration ID: 32880).

### Data analysis

The primary analysis will use a generalised linear mixed model with a logit link and a random effect for clinic, allowing for the clustering of participants within clinics. The model will control for the clinic-level baseline antibiotic prescribing rate as well as potential confounders, and the full list of model covariates will be prespecified and set out in the statistical analysis plan. Results will be reported as ORs with 95% CIs.

Missing data will be treated as missing at random and imputed using a chained-equations multiple imputation

**Table 1** List of the primary and secondary outcome measures and when they will be collected at follow-up audit

| Item | Outcome measure | Initial consultation | Patient telephone interview after consultation | | |
| | | | 7 days | 14 days | 21 days |
|---|---|---|---|---|---|
| Primary outcome | Prescribing rate of antibiotics (%) | ✓ | | | |
| Secondary outcomes | Patient satisfaction | ✓ | | | |
| | Patient symptom severity and duration | | ✓ | ✓ | ✓ |
| | Patient recovery | | ✓ | ✓ | |
| | Patient consumption of antibiotics | | ✓ | ✓ | ✓ |
| | Patient attitude, beliefs and understanding of the patient leaflet | | ✓ | | |
| | Patient costs | | | | ✓ |
| | Patient quality of life (EQ-5D-5L*) | ✓ | ✓ | ✓ | ✓ |

EQ-5D-5L refers to a standardised measure of health-related quality of life developed by the EuroQol Group to provide a simple, generic questionnaire for use in clinical and economic appraisal or population health status surveys. The descriptive system comprises five dimensions: mobility, self-care, usual activities, pain/discomfort and anxiety/depression. Each dimension has 5 levels: no problems, slight problems, moderate problems, severe problems and extreme problems.

model. A complete-cases analysis will be presented as a secondary analysis.

The analysis of secondary outcomes will use a similar modelling framework with distributions appropriate to the outcome, for example, linear modelling for continuous outcomes and Poisson/negative binomial for count outcomes. These models will control for clustering and the factors listed above.

All results will be reported in line with the Cluster Randomised Trials extension to the CONSORT (Consolidated Standards of Reporting Trials) guidelines. No interim analyses are planned. Full details of the analyses to be undertaken will be set out in the statistical analysis plan and approved by the Trial Steering Committee (TSC).

### Sample size calculation

Our previous study recorded an antibiotic prescribing rate of 75% for rural township health centres in Anhui province[21 28] and a previous pilot intervention showed a reduction in antibiotic prescribing of more than 20%.[27] There is no internationally agreed minimally important difference in antibiotic prescribing, however, we anticipate that a reduction of at least 15% in antibiotic prescribing would represent a meaningful change at the population level given the very high rates of antibiotic prescribing.

To detect a possible absolute difference of 15% in prescribing rates between intervention (60%) and control practices (75%) with 90% power and alpha 0.05, we will need 203 patient consultations for infection RTI in each arm. Based on our previous study results the estimated ICC value was 0.05 and a design effect value would be 2.45. We would, therefore, aim to recruit at least 2.45*203*2=995 patients into the study. Allowing for a loss to follow-up rate of 10% and a cluster size of 30 patients per township health

centre, we will recruit 40 township health centres into the study and each arm will have 20 township health centres.

### Economic evaluation

The economic evaluation will be taken from a societal perspective and consider cost implications for patients as well as health services. Resource usage information will be costed using corresponding Chinese tariff.

All itemised resource usage data will be costed using appropriate data, with unit costs of visiting healthcare facilities based on medical service price items. Medication based on clinical consultation patient questionnaire and phone interviews with the patients at 7, 14 and 21 days; and the time spent costed at average wage in Anhui province. Insurance coverage will be weighted to calculate private spending.

Resource use will be weighted by its unit cost over the study period. Accumulated costs per patient will be calculated for the study period. Quality of life data will be translated into a utility score using a Chinese tariff. Quality-adjusted life-years (QALYs) will be estimated using the area under the curve approach.

Missing data will be handled by multiple imputation methods. Difference in costs and QALYs will be compared using a generalised linear mixed model taking clinic as cluster level and by adjusting baseline clinical and quality of life variables. Bootstrapping methods will be used to estimate the incremental cost per QALYs gained and per percentage reduction of antibiotic prescribing. Major assumptions will be explored through sensitivity analyses. Completed cases analyses will be presented as one of such scenario analyses.

### Qualitative process evaluation

A Mandarin-speaking qualitative researcher will conduct observations of the training process and intervention

implementation. The key content related to participants during the training process, including any additions by the trainer, questions and answers, comments or discussion among participants and non-verbal reactions displayed by participants will be recorded. The researcher will visit intervention clinics and observe how the intervention is being implemented.

A Mandarin-speaking qualitative researcher will conduct semistructured interviews with the intervention practitioners. Short interviews will be conducted at the end of the training, to elicit intervention practitioners' views about the training. After intervention practitioners have had time to implement the intervention in their clinics, they will be invited to participate in a second semistructured interview to understand their experiences of the intervention. They will be asked to describe how they used the intervention in their practice, any barriers and facilitators encountered in following the intervention and whether it influenced their views or practice.

A purposive sample of up to 40 patients who are recruited into the study will be interviewed. Patient will be selected to capture a range of sociodemographic characteristics and illness presentations. Some will be interviewed immediately after completing their consultation with health practitioners and some after they have completed the telephone questionnaires. Interviews will be semistructured and guided by a topic guide that will cover experience of the consultation and understandings of the patient leaflet they received.

All semistructured interviews will be transcribed. The qualitative data will be analysed thematically.

### Data handling and repository

This trial will be monitored by a TSC and collected data will be monitored by data monitoring committee. There are no anticipated risks to patients from taking part in the study and participants have the right to withdraw from the study at any time without giving a reason.

Hard copies of documents with identifiable details of patient, practitioner and clinic will be stored in lockable cabinets on premises approved by the principal investigator or relevant co-investigators at Anhui Medical University and any data created will be encrypted and stored on University secure servers until they can be uploaded or scanned and stored in the University of Bristol Research Data Storage Facility (RDSF) at the first available opportunity. Only authorised users can access data stored within the RDSF and procedures will be put in place to allow authenticated, external collaborators to view add and/or edit data.

When possible, data will be exported directly to databases and consistency maintained via peer review and cross-checks with previous measurements. Manually inputted data will be double entered. Quality of questionnaire and interview data will be assured by training, provision of an interview manual and resurvey of 5% of subjects.

### Patients and public involvement

Patients (and practitioners) contributed to the co-design of the intervention. They also provided feedback on piloted data collection forms, which led to modifications. The results will be disseminated to study participants and other stakeholders. At the end of the research project we will hold a high-level meeting to disseminate our findings to key stakeholders and develop a strategy for impact with relevant stakeholders.

## ETHICS AND DISSEMINATION

Ethical approval was obtained from the Biomedical Research Ethics Committee of Anhui Medical University (Ref: 20180259); the study has undergone due diligence checks and is registered at the University of Bristol (Case No: 2020-3137). This project is one of the three work packages in the UK-China AMR Centre Partnerships 2018 (Strategies to reduce the burden of antibiotic resistance in China) supported by UKRI (UK Research and Innovation) Medical Research Council (Grant Ref: MR/S013717/1)) and National Natural Science Foundation of China (Grant Ref: 81861138049). The protocol is rooted in a multidisciplinary collaboration between institutions in the UK and China.

Throughout the research project we will involve key stakeholders with roles in managing antibiotic use and AMS. Research findings will be translated into short policy briefings with recommendations informed by existing policies in China and to influence current practice in healthcare settings. The credibility of research findings will be indicated to stakeholders through peer-reviewed publications in internationally recognised journals and presentations at academic conferences in China, the UK and internationally.

**Author affiliations**
[1]Population Health Sciences, University of Bristol, Bristol, UK
[2]School of Health Services Management, Anhui Medical University, Hefei, Anhui, China
[3]School of Psychological Sciences, University of Bristol, Bristol, UK
[4]University of Leicester Department of Health Sciences, Leicester, UK
[5]Primary Care and Population Science, University of Southampton, Southampton, UK
[6]Department of Epidemiology and Biostatistics, Anhui Medical University, Hefei, Anhui, China
[7]National Infection Service, Public Health England South Region, Bristol, UK
[8]Institute of Clinical Pharmacology, Peking University First Hospital, Beijing, China

**Contributors** HL, DW, CC and LY were responsible for developing the research questions and HL, DW, CC, LY, PL, BS, GLY, IO and BZ were responsible for the study design. TZ, JCha, JChe, XS, RL, XH, Y-HS are responsible for data collection and WC, JC and LZ are responsible for study management and coordination. WC, JC, BS, TZ, GLY and CC drafted the paper. All authors read, commented on and approved the final manuscript.

**Funding** The authors acknowledge MRC and Newton Fund through a UK-China AMR Partnership Hub award (MR/S013717/1), and National Natural Science Foundation of China award (81861138049).

**Competing interests** None declared.

**Patient consent for publication** Not applicable.

**Provenance and peer review** Not commissioned; externally peer reviewed.

**Open access** This is an open access article distributed in accordance with the Creative Commons Attribution 4.0 Unported (CC BY 4.0) license, which permits others to copy, redistribute, remix, transform and build upon this work for any purpose, provided the original work is properly cited, a link to the licence is given, and indication of whether changes were made. See: https://creativecommons.org/licenses/by/4.0/.

**ORCID iDs**
Wenjuan Cong http://orcid.org/0000-0002-5034-8182
Jing Chai http://orcid.org/0000-0003-3770-2279
Tingting Zhang http://orcid.org/0000-0002-2612-0014
Paul Little http://orcid.org/0000-0003-3664-1873
DeBin Wang http://orcid.org/0000-0002-9708-9659

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
