## [Reviewer comments · BMJ Open]

ARTICLE DETAILS

TITLE (PROVISIONAL)	A cluster randomised controlled trial to assess a tailored intervention to reduce antibiotic prescribing in rural China – Study protocol
AUTHORS	Cong, Wenjuan; Chai, Jing; Zhao, Linhai; Cabral, Christie; Yardley, Lucy; Yao, Guiqing; Zhang, Tingting; Cheng, Jing; Shen, XingRong; Liu, Rong; Little, Paul; Stuart, Beth; Hu, Xiaowen; Sun, Ye-Huan; Oliver, Isabel; Zheng, Bo; Lambert, Helen; Wang, DeBin

VERSION 1 – REVIEW

REVIEWER	Qiang Sun Shandong University
REVIEW RETURNED	07-Feb-2021

GENERAL COMMENTS	The topic is very interesting, but several things related to the intervention should be further explained.  1. Please describe the feasibility and generalizability of the complex intervention 2. It is better to provide a map to illustrate the sampling 3. The peer support design is good, but please explain which methods to be used to ensure them to follow the requirements of the intervention. 4. The compute-based aid system is very important in the intervention, but this manuscript just give us very general information, please illustrate it in detail. 5. Some supplementary documents are not final version, please check it. 6. Please justify why conduct the economic evaluation 7. Please justify the reasonability of doing utility measurement.
--

REVIEWER	Emil Lesho Rochester Regional Health
REVIEW RETURNED	11-Mar-2021

GENERAL COMMENTS	 1) If the intervention involves "a computer-based decision support system" why are authors excluding practitioners "who have basic computer skills" In rural villages, could not this end up excluding a significant no. of providers? 2) How will authors avoid recruitment / selection bias? In other words, how will they ensure a fair, balanced, and representative group of practitioners are selected / included? 3) Why are the authors excluding so many types of patients? The groups they are excluding are some of the very groups in which antibiotics are over prescribed, namely patients without bacterial infections, immunocompromised, patient who previously sought treatment for an RTI, and patients w/ non-infective conditions. This
--

	could artificially increase the effect of the intervention. 4) How will prescribing rate (or in what units) be measured? Same question for patient consumption. 5) In my opinion, patient consumption should be evaluated at every interview , not just on day 21,
--	--

VERSION 1 – AUTHOR RESPONSE

Reviewer: 1

Prof. Qiang Sun, Shandong University

Comments to the Author:

The topic is very interesting, but several things related to the intervention should be further explained.

1. Please describe the feasibility and generalizability of the complex intervention

Reply: Thank you for your excellent question. Before launching this project, our authors (DW, HL, IO, JC, LZ et al.,) had worked together to investigate patterns of antibiotic treatment-seeking, extents of and drivers for antibiotic use for common infections, and also test the feasibility of determining the prevalence and epidemiology of AMR in rural areas of Anhui Province. The previous project has laid solid foundations for us to develop intervention tools (e.g., computer-based decision support system, training materials for village practitioners) and data collection forms for this trial (see previous papers: 1) Zhao L et al., Pathways to optimising antibiotic use in rural China: identifying key determinants in community and clinical settings, a mixed methods study protocol. *BMJ Open* 2019;9:e027819; 2) Shen X et al., Web-Based Just-in-Time Information and Feedback on Antibiotic Use for Village Doctors in Rural Anhui, China: Randomized Controlled Trial. *J Med Internet Res.* 2018;20(2):e53; 3) Chai J et al., Cross-sectional study of the use of antimicrobials following common infections by rural residents in Anhui, China. *BMJ Open* 2019;9:e024856; 4) Kwiatkowska, R et al., Patients without records and records without patients: Review of patient records in primary care and implications for surveillance of antibiotic prescribing in rural China; *BMC Health Services Research.* 2020; 20: 564-572; 5) Chen M et al., Prescribing antibiotics in rural China: the influence of capital on clinical realities. *Front. Sociol.* 2020, 5: 66-75).

We optimised the intervention, using qualitative research to identify barriers to acceptability, feasibility and engagement that need to be addressed. We had conducted 4 rounds of ‘think-aloud’ interviews with 37 village practitioners and 17 patients in rural Anhui. Think-aloud interviews obtain detailed user views by asking participants (village practitioners and patients) to view contents of our intervention tools (training materials for village practitioners, computer-based decision support system, patient leaflet, and peer support system). Findings were immediately tabulated and analysed to identify changes required, modified materials used for the next round of think-aloud interviews and the process repeated until participant feedback indicates no further important, feasible and appropriate changes are needed. Through this process, we created a feasible and generalisable intervention to reduce antibiotic prescribing tailored for the rural Chinese context. We have also added this text to the Intervention section (see Page 5).

About data collections forms, we did a pilot study and invited village patients and practitioners to provide comments and feedbacks on those forms; and we have further modified some of the questions accordingly.

2. It is better to provide a map to illustrate the sampling

Reply: Thank you for your question. We will recruit 40 township health centres from 10 counties (4 township health centres per county), those counties will be randomly selected from 4 cities

geographically dispersed across Anhui Province. Specifically, We will randomly recruit 6 counties from 2 cities in densely populated North Anhui and 4 counties from another 2 cities from sparsely populated South Anhui. We will seek support from local county health authorities during the recruitment of township health centres. They will provide a list of townships in their area and the research team will randomly select 4 township health centres.

As stated, this is a randomised recruitment and currently the recruitment is still ongoing, at this stage we could not be able to provide a map to illustrate the sampling. We could provide a sampling map once the recruitment is complete.

3. The peer support design is good, but please explain which methods to be used to ensure them to follow the requirements of the intervention.

Reply: Thank you for this useful question. Once we set up Wechat peer support groups for health practitioners from the intervention group and one large technical support group for all village practitioners in the intervention group, as an on-line forum for general queries or trouble shooting; we will identify the peer group leads/champions to organize and lead the peer group audit and feedback discussions and we will provide a meeting topic guide for their virtual group discussion. During the group discussion, member of the group will share their experience of applying the intervention, discuss any problems and support each other to find solutions.

To ensure them to follow the requirements of the intervention, a senior clinician (Dr Xiaowen Hu, a co-author for this manuscript) will be the senior clinical champion for intervention and of the member of all peer support groups and the technical support group. He will oversee the group discussion and make sure it complies with the intervention format for audit and feedback. Additionally, field researchers, will also join those Wechat subgroups as observers; and they will also record the content of all group discussions and anonymize all transcripts for data analysis. We have amended the text to the Intervention section (see page 6)

4. The compute-based aid system is very important in the intervention, but this manuscript just give us very general information, please illustrate it in detail.

Reply: Thank you for this useful suggestion. We have revised the text to the Intervention section (see page 6). The computer-based aid system is a decision support system and also clinic data collection system for village practitioners to use during the consultation. It has been adapted from our previous project (Shen X et al., Web-Based Just-in-Time Information and Feedback on Antibiotic Use for Village Doctors in Rural Anhui, China: Randomized Controlled Trial. *J Med Internet Res.* 2018;20(2):e53)). Generally, it has the following sections: 1) eligibility criteria: help practitioners recruit suitable patients for this study; 2) patient symptoms: help practitioners identify what symptoms patients are suffering from; 3) diagnosis, patient reassurance & prognosis: assist practitioners make diagnosis, and provide appropriate patient reassurance and prognosis for each type of RTI patient; 4) treatment recommendations: provide very detailed treatment recommendations for each type of RTI patient; 5) Table of Symptomatic treatment options for RTI patients (from China National Guidelines); 6) Safety netting advice (when to come back to see the doctor). We have also provided this doc as an appendix in the appendments.

5. Some supplementary documents are not final version, please check it.

Reply: Thank you for pointing out this, which we have corrected (please see the uploaded final version of supplementary docs).

6. Please justify why conduct the economic evaluation; 7, Please justify the reasonability of doing utility measurement.

Reply: Thank you for your useful comments. As we stated in the Introduction section that, in addition to the implications for AMR caused by inappropriate use of antibiotics, there is a direct cost to village patients for extra health resource utilization (consultation fees, medication costs, loss of productivity costs etc) and most of which would not be reimbursed through national medical scheme i.e. China's Rural Cooperative Medical Scheme. Additionally, there are cost implications to the health services, no matter whether patient treatment costs would be covered partially by medical insurance scheme or mostly by out-of-pocket spending from patients themselves. Economic evaluation would be very useful for this project, as we will not only investigate whether our intervention would be effective in reducing antibiotic prescribing in rural China; and we could also estimate costs and cost-effectiveness of this intervention on antibiotic prescribing. Doing utility measurement with corresponding Chinese tariffs enables us to quantify our economic evaluation outcomes for example, we could do incremental costs and cost-effectiveness comparison between intervention and control arms in this trial.

This project is one of the three work packages in the UK-China AMR Centre Partnerships 2018 (Strategies to reduce the burden of antibiotic resistance in China) supported by UKRI Medical Research Council (Grant Ref: MR/S013717/1). We have another two work packages: WP3 - Estimate socio-economic burden of ABR in China and evaluate the cost effectiveness of different intervention strategies; WP2- Measure community exposure to antibiotics from direct and indirect (water and food animal) sources and resulting ABR using wastewater epidemiology, microbiological analysis and genome sequencing. The economic evaluation work of this project would also provide data for modelling for WP3 of our programme.

Reviewer: 2

Dr. Emil Lesho, Rochester Regional Health

Comments to the Author:

1) If the intervention involves "a computer-based decision support system" why are authors excluding practitioners "who have basic computer skills" In rural villages, could not this end up excluding a significant no. of providers?

Reply: Thank you for pointing out this error, we really appreciate it. It should read "health practitioners who don't have basic computer skills". We do need to recruit practitioners with basic computer skills and could use our computer-based decision support system in their practice. We have corrected this in the manuscript (see page 4, section "recruitment of practitioners").

2) How will authors avoid recruitment / selection bias? In other words, how will they ensure a fair, balanced, and representative group of practitioners are selected / included?

Reply: Thank you for this helpful comment. In rural China, we will approach township health centres for recruitment with the support of local county health authorities. Once township health centres agree to participate in our study, we will then invite all their eligible practitioners to participate in our trial.

We will recruit a balanced and representative sample of township health centres for this study. Specifically, we will randomly recruit 40 township health centres from 10 counties (4 township health centres per county), within 4 regions across the whole Anhui Province.

The recruitment of health practitioners will be fair. Once a township health centre is recruited and has given permission for its staff to participate, all eligible practitioners working in that health centre are likely to agree to participate in the study. We will only exclude practitioners who either are not involved in clinical diagnosis and medicine prescribing; or do not have basic computer skills as we need health practitioners to use our computer-based decision support system if they are allocated to the intervention group. We have also amended the text in the section "recruitment of practitioners" (see page 4).

3) Why are the authors excluding so many types of patients? The groups they are excluding are some of the very groups in which antibiotics are over prescribed, namely patients without bacterial infections, immunocompromised, patient who previously sought treatment for an RTI, and patients w/ non-infective conditions. This could artificially increase the effect of the intervention.

Reply: Thank you for your questions. Patients without bacterial infections, immunocompromised patients and patients with non-infective conditions are not excluded from this study if they present with any of the symptoms listed in the inclusion criteria. Patients who previously sought treatment for the same episode of illness in other health services (outpatient clinic in the county hospitals etc) will be excluded, simply because the treatment sought elsewhere may bring interference or contamination to our study findings by having previously obtained antibiotics elsewhere. We have amended the text in the section “Exclusion criteria” (see page 5).

4) How will prescribing rate (or in what units) be measured? Same question for patient consumption.

Reply: Thank you for your questions. Antibiotic prescribing rate (%) will be measured by dividing the total number of RTI patients who are prescribed antibiotics (IV and oral) against the total number of RTI patients who are present to each township health centre. During the initial consultation, any antibiotics or other medicines prescribed and mode of administration (IV or oral), name, dosage and quantity (days) for recruited patients will be recorded by a field researcher on site. In the follow-up telephone interview on Day 7, 14 and 21, our field researcher will check with patients the actual consumption of antibiotics (how many days they have taken the antibiotics, dosage etc). Through this, we could estimate patient adherence to follow village practitioner’s advice (%) by dividing the actual patient consumption quantity of antibiotics against the prescribed quantity of antibiotics. We apologise that we did not make this clear in the manuscript and we have changed it accordingly (see revised text in Section “Randomisation, allocation & blinding (page 5) and also in section “Data collection methods” (page 7).

5) In my opinion, patient consumption should be evaluated at every interview , not just on day 21,

Reply: Thank you for your useful suggestion and glad that we share the same thoughts on this. Patient antibiotics consumption information was planning to be only collected on Day 21 after initial consultation, when we submitted our manuscript. Later we also realised the importance to evaluate patient antibiotics consumption at different follow-ups; so we will collect this information at every interview. We have amended the text in the section “Data collection methods” (see page 7) and Table 1 (see page 12).

VERSION 2 – REVIEW

REVIEWER	Qiang Sun Shandong University
REVIEW RETURNED	27-Aug-2021

GENERAL COMMENTS	I recognized that authors have greatly improved the manuscript. However, I would like them to further improve the following points:  1. Line 36-42, page 7. The authors only collected 10 patients’ information for each township health centre in baseline audit, it’s not clear whether representative enough to determine high or low antibiotic prescription rates in township health centers, please illustrate it in detail. 2. Line 10-28, page 9. Senior clinicians are often burdened with heavy workload. Please illustrate which method to be used to ensure them plays an oversee role in peer support?
--

	3. Line 29-40, page 9. Please give us more details on the working mechanism of computer-based decision support system intervention, such as how to help practitioners identify what symptoms patients are suffering from and make a diagnosis. 4. Line 14-17, page 10. Follow-up is quite important and lots of contents need to be recorded in this study. Please explain which method to be used for quality control during the three times telephone interviews follow-up, such as how to ensure participants will not be lost? 5. Please improve the description of feasibility and sustainability in this manuscript. 6. Is there a pilot intervention? It is crucial for the validation of feasibility for the complex interventions.
--	---

VERSION 2 – AUTHOR RESPONSE

Reviewer: 1

Prof. Qiang Sun, Shandong University

Comments to the Author:

I recognized that authors have greatly improved the manuscript. However, I would like them to further improve the following points:

1. Line 36-42, page 7. The authors only collected 10 patients' information for each township health centre in baseline audit, it's not clear whether representative enough to determine high or low antibiotic prescription rates in township health centers, please illustrate it in detail.

Reply: We acknowledge that 10 patients per health centre is not ideal. This was a pragmatic decision to accommodate the reduced timeline for the audit that resulted from Covid pandemic related interruptions to the study. However, the average prescribing rate across the clinics is 73%, which is the rate we expected based on a previous larger study. The audit data will enable us to identify roughly half of the health centres as above average and roughly half as below average in terms of their antibiotic prescribing rate. Since there is no other reliable measure of antibiotic prescribing rates at a health centre level, obtaining some baseline audit data will maximise the likelihood that higher and lower prescribing practices will be divided equally between the arms.

2. Line 10-28, page 9. Senior clinicians are often burdened with heavy workload. Please illustrate which method to be used to ensure them plays an oversee role in peer support?

Reply: We accept that senior clinicians are often burdened with heavy workload. However, the senior clinician will not be providing any clinical advice and will not be in any peer support group. Peer support group membership is restricted to intervention arm clinicians. The senior clinician and other members of the trial team will be part of the large technical support group where all intervention practitioners can post questions about the intervention (but not clinical cases). The senior clinician will be able to provide general guidance and reassurance that the intervention follows National Chinese prescribing guidance but will not provide specific clinical advice on cases. The senior clinician also has two assistants (junior clinicians) who will also help oversee the technical support group discussion

and feedbacks from the peer support group leader, and make sure it complies with the intervention format for audit and feedback.

We apologize for the confusion and ambiguity in this section, and we have revised it appropriately (see Paragraph 4, Section “Intervention”)

3. Line 29-40, page 9. Please give us more details on the working mechanism of computer-based decision support system intervention, such as how to help practitioners identify what symptoms patients are suffering from and make a diagnosis.

Reply: Thank you for your question. The detail of content of computer-based decision support system is in Appendix 5.

First, we would like to clarify that practitioners will make their own independent judgements on patient symptoms, diagnosis and treatment decisions. In the DSS system, practitioners record their diagnosis by selecting from the different acute respiratory tract infections listed in the Chinese National Guidelines. The DSS will then provide a summary of the treatment recommendations for the selected respiratory infection from the Chinese National Guidelines. The DSS system also provides advice on patient reassurance, prognosis and safety netting and symptomatic treatment options for acute respiratory tract infections (which all complies with the Chinese National Guidelines). The computer-based decision support system is mainly to provide treatment recommendations from the Chinese National Guidelines for Treating Acute Respiratory Tract Infections, and also help practitioners to treat RCT patients with fewer antibiotic prescriptions more confidently by following the Chinese National Guidelines.

We have now made this clearer in the manuscript (see Paragraph 5, Section “Intervention”).

4. Line 14-17, page 10. Follow-up is quite important and lots of contents need to be recorded in this study. Please explain which method to be used for quality control during the three times telephone interviews follow-up, such as how to ensure participants will not be lost?

Reply: In pilot work, we were able to achieve complete data for the follow up interviews at day 7, 14 and 21 for 86%, 72% and 62% of patients (respectively). The lower response rate on day 21 was mainly due to the interruption of data collection by the Covid pandemic. To ensure we can have a satisfactory follow-up rate for telephone interviews on Day 7, 14 and 21 after the initial consultation, patients will be asked to provide their contact information (their phone number, their family’s phone number as a back-up, and also their address in case we cannot contact them by calls), when they agree to participate in our trial. Additionally, our field researchers will schedule telephone interview slots with patients in advance.

Regarding quality control for follow-up data collection, when possible, follow-up telephone interview data will be exported directly to databases. Data consistency and cross-checks with previous measurements will be reviewed throughout the trial period.

5. Please improve the description of feasibility and sustainability in this manuscript.

Reply: We apologise for ambiguities in the description of the our assessment of the feasibility of the peer support group and computer-based decision support system. We have revised the description of intervention thoroughly (please see Section “Intervention”)

6. Is there a pilot intervention? It is crucial for the validation of feasibility for the complex interventions.

Reply: We were not able to carry out a pilot intervention. However, as we mentioned in the manuscript, practitioners and patients were involved in the co-design of the intervention tools: they participated in several rounds of think-aloud interviews by providing their views, comments and feedbacks on our intervention tools; and helped us finalise intervention tools (see detail Paragraph 1 & 2, section “Intervention”). The process of “think-aloud” interviews with village practitioners and patients has not only helped co-design the intervention tools, but also laid a solid basis to enable the intervention tools to be feasible and generalisable in the trial settings.

As an important element of the intervention, our computer-based decision support system is adapted from a previous project where we set up a web-based just-in-time information system, and our colleagues had already tested the feasibility and acceptance of the system with a pilot study (Shen X et al., Web-Based Just-in-Time Information and Feedback on Antibiotic Use for Village Doctors in Rural Anhui, China: Randomized Controlled Trial. *J Med Internet Res.* 2018;20(2):e53).

Additionally, we organised several trial steering group meetings where the intervention tools and their feasibility and generalisability were intensively discussed before we launched the trial.